# High-content high-resolution microscopy and deep learning-assisted analysis reveals host and bacterial heterogeneity during *Shigella* infection

Ana Teresa López-Jiménez*, Dominik Brokatzky, Kamla Pillay, Tyrese Williams, Gizem Özbaykal Güler, Serge Mostowy*

Department of Infection Biology, London School of Hygiene and Tropical Medicine, London, United Kingdom

## eLife Assessment

This manuscript describes an AI-automated microscopy-based approach to characterize both bacterial and host cell responses associated with *Shigella* infection of epithelial cells. The methodology is **compelling** and should be helpful for investigators studying a variety of intracellular pathogens. The authors have acquired **important** findings regarding host and bacterial responses in the context of infection, which should be followed up with further mechanistic-based studies.

**\*For correspondence:**
ana.lopezjimenez@lshtm.ac.uk
(ATL-J);
serge.mostowy@lshtm.ac.uk
(SM)

**Competing interest:** The authors declare that no competing interests exist.

## Abstract
*Shigella flexneri* is a Gram-negative bacterial pathogen and causative agent of bacillary dysentery. *S. flexneri* is closely related to *Escherichia coli* but harbours a virulence plasmid that encodes a type III secretion system (T3SS) required for host cell invasion. Widely recognised as a paradigm for research in cellular microbiology, *S. flexneri* has emerged as important to study mechanisms of cell-autonomous immunity, including septin cage entrapment. Here, we use high-content high-resolution microscopy to monitor the dynamic and heterogeneous *S. flexneri* infection process by assessing multiple host and bacterial parameters (DNA replication, protein translation, T3SS activity). In the case of infected host cells, we report a reduction in DNA and protein synthesis together with morphological changes that suggest *S. flexneri* can induce cell-cycle arrest. We developed an artificial intelligence image analysis approach using convolutional neural networks to reliably quantify, in an automated and unbiased manner, the recruitment of SEPT7 to intracellular bacteria. We discover that heterogeneous SEPT7 assemblies are recruited to bacteria with increased T3SS activation. Our automated microscopy workflow is useful to illuminate diverse host and bacterial interactions at the single-cell and population level, and to fully characterise the intracellular microenvironment controlling the *S. flexneri* infection process.

## Introduction

*Shigella flexneri* is a Gram-negative bacterium and the aetiologic agent of bacillary dysentery, an intestinal infectious disease that kills more than 150,000 people worldwide per year, disproportionally affecting children under 5 years of age from low- and middle-income countries (*Global Burden of Disease Collaborative Network, 2020*). After the ingestion of a small number of bacteria (10–100), *S. flexneri* rapidly colonises the human intestinal epithelium. For this, *S. flexneri* uses a type III secretion system (T3SS) that injects bacterial effectors into the host cell to modulate key aspects of the infection process and to evade innate immune defences by the host (*Schnupf and Sansonetti, 2019*).

Following uptake by host cells, *S. flexneri* breaks its vacuolar compartment to access the cytosol (*Ray et al., 2010*). Here, the virulence factor IcsA enables *S. flexneri* to hijack host actin and polymerise comet tails that render bacteria motile to disseminate to neighbouring cells and spread in the tissue (*Bernardini et al., 1989*; *Egile et al., 1999*).

Septins are highly conserved GTP-binding proteins that form heteromeric complexes (hexamers, octamers) that assemble into non-polar filaments, bundles, and rings (*Martins et al., 2023*; *Soroor et al., 2021*; *Szuba et al., 2021*; *Woods and Gladfelter, 2021*). Septins are involved in many physiological cellular processes (e.g. cytokinesis, cilia formation) and play key roles during infection (*Mostowy and Cossart, 2012*; *Robertin and Mostowy, 2020*; *Spiliotis and Nakos, 2021*). Septins form collar-like structures at the plasma membrane during bacterial invasion (*Boddy et al., 2018*; *Mostowy et al., 2009*; *Robertin et al., 2023*) and ring-like structures surrounding membrane protrusions in cell-to-cell bacterial spread (*Mostowy et al., 2010*). In the cytosol, septins are recruited to actin-polymerising bacteria, including *S. flexneri* and *Mycobacterium marinum*, as cage-like structures (*Mostowy et al., 2010*). In the case of *S. flexneri*, septins are recruited to μm-scale curvature at bacterial poles and the division site, which are enriched in the negatively charged phospholipid cardiolipin (*Krokowski et al., 2018*). Recent work has shown that IcsA promotes the recruitment of septins to *S. flexneri*, and that lipopolysaccharide inhibits *S. flexneri*-septin cage entrapment (*Lobato-Márquez et al., 2021*; *Mostowy et al., 2010*). Septin cages prevent bacterial actin tail motility and therefore are considered as an antibacterial mechanism of host defence (*Krokowski et al., 2018*; *Mostowy et al., 2010*).

Intracellular infection is usually portrayed as a uniform, synchronised, and highly orchestrated process. However, infection assays performed in controlled laboratory environments with isogenic populations display a high level of complexity (*Avraham et al., 2015*; *Toniolo et al., 2021*). It is particularly unknown how different subsets of bacteria (e.g. motile vs non-motile bacteria) can colonise distinct intracellular microenvironments and become targeted by multiple cell-autonomous immunity mechanisms (e.g. septin cages, autophagy). It has been proposed that this diversity of intracellular niches impacts bacterial physiology (*Day et al., 2024*; *Gutierrez and Enninga, 2022*; *Santucci et al., 2021*). For example, it may affect individual bacterial ability to acquire nutrients, to be exposed to varying antibiotic concentrations, or to elicit different responses and signalling pathways within the host cell. While the concept of phenotypic heterogeneity in prokaryotic clonal populations is well established (*Ackermann, 2015*), the concept is also emerging in eukaryotic cells where non-genetic differences may, for instance, explain individual cell sensitivity and responses by cell-autonomous immunity (*McDonald and Dedhar, 2024*). In addition, previous high-content studies coupled with mathematical modelling have shown that host cell context (including local cell density, lipid composition at the plasma membrane, or previous infection status) affects their susceptibility to viral and bacterial infection (*Snijder et al., 2009*; *Voznica et al., 2018*). It is therefore paramount to study host and bacterial cells at the single-cell level, to assess the individual responses that collectively contribute to the infection process.

Here, to dissect the complexity of *S. flexneri* infection in epithelial cells, we use high-content high-resolution microscopy coupled to automated image analysis. High-content microscopy enables the automated acquisition of thousands of microscopy images and therefore is useful to capture the heterogeneity of individual host and bacterial cells during the infection process for quantitative analysis (*Aylan et al., 2023*; *Brodin and Christophe, 2011*; *Deboosere et al., 2021*; *Dramé et al., 2023*; *Fisch et al., 2019*; *Lensen et al., 2023*; *Pylkkö et al., 2021*). We analyse multiple parameters, such as morphological host cell features, the de novo synthesis of DNA, and proteins in both host and bacterial cells, as well as activation of the bacterial T3SS. To investigate the antimicrobial potential of septin-mediated immunity, we develop a deep learning-based approach to automatically identify septin-*S. flexneri* interactions. Together, we highlight the power of high-content microscopy and artificial intelligence to reveal insights on both host and bacterial cells and illuminate fundamental biology underlying the *S. flexneri* infection process.

## Results

### Capturing the heterogeneity of *S. flexneri* infection using high-content microscopy

To understand the complexity and reveal the heterogeneity of *S. flexneri* infection in human epithelial cells, we dissected the infection process using high-content high-resolution microscopy and HeLa cell infection model (*Figure 1A and B*, *Figure 1—figure supplement 1A*). We observed a mean of 11.5% of infected cells measured by fluorescence at 3 hr 40 min post infection (hpi) at multiplicity of infection (MOI) 100:1 (*Figure 1C*). These infected cells presented different burdens of bacteria following an exponential decay distribution (*Figure 1D*), reflective of the formation of infection foci (*Ortega et al., 2019*). With growing evidence of the cytosol being a non-homogeneous compartment containing concentration gradients, biomolecular condensates, protein complexes, and concentrates (*van Tartwijk and Kaminski, 2022*), we tested whether *S. flexneri* may be exposed to different intracellular microenvironments. For this, we measured the distance of bacteria to the centroid of cells (*Figure 1E*) and observed a median distance of 8.74 µm. The median distance to the centroid of the cell significantly decreased with increasing bacterial burden (*Figure 1E*), suggesting that the microenvironment of bacteria is diverse and changes as bacteria replicate intracellularly and populate the central parts of host cytosol after uptake at the host cell periphery. To dissect the heterogeneity in host cells, we quantified an increase of 36.9% and 22.3% in the cellular and nuclear area of infected cells, respectively (*Figure 1G and I*), which was dependent on the infection burden (*Figure 1H and J*). In addition, the total amount of Hoechst fluorescence in the cell nucleus increased 29.4% in infected cells (*Figure 1—figure supplement 1B*), indicating either an increase in nuclear volume or altered chromatin condensation. To test if this effect was due to morphological rearrangements during infection, we treated cells with latrunculin B to inhibit actin polymerisation. While cells treated with latrunculin B decreased their cellular area (but not their nuclear area) (*Figure 1—figure supplement 1C and D*), the drug did not abrogate the difference in size between infected and uninfected cells. Taken together, these results highlight profound morphological rearrangements during infection that are dependent on the infection burden.

### Epithelial cells reduce de novo DNA and protein synthesis upon infection

In order to understand the physiological state of enlarged epithelial cells infected with *S. flexneri*, we monitored host DNA replication and protein translation using Click chemistry. Using a 1 hr pulse of the clickable thymidine analogue EdU, we specifically visualised and quantified de novo DNA synthesis in the population of epithelial cells with high-content microscopy (*Figure 2A*; *Figure 2—figure supplement 1*). We identified the subpopulation of cells in S-phase by their high level of EdU incorporation. Consistent with this, this subpopulation of cells was fully abrogated in the presence of the S-phase blocker aphidicolin (*Pedrali-Noy et al., 1980*; *Figure 2—figure supplement 1A and B*). Approximately one third of uninfected cells (34.45%) were found in S-phase (*Figure 2B and C*), in agreement with S-phase lasting around one third of the cell cycle (~8 hr out of ~22 hr). Infected cells presented a similar proportion of cells in S-phase (35.89%), indicating that *S. flexneri* infection did not block transition of cells from G1 to S at the timepoint tested. We then assessed the amount of de novo DNA synthesis incorporated in cells in S-phase during infection. Surprisingly, we observed a reduction of EdU incorporation in infected cells (*Figure 2D*), which was dependent on the bacterial burden (*Figure 2E*). These results indicate that *S. flexneri* infection slows down DNA synthesis of epithelial cells.

To assess protein translation, we used a 1 hr pulse of the clickable methionine analogue AHA, which specifically labels de novo synthesized proteins as AHA incorporation is reduced after treatment with the inhibitor of translational elongation cycloheximide (*Ennis and Lubin, 1964*; *Figure 2—figure supplement 1C and D*). During infection, we observed a reduction of protein translation in infected cells compared to non-infected cells (*Figure 2F and G*), which was also dependent on the bacterial burden (*Figure 2H*).

Collectively, the cell morphological rearrangements, together with the reduction of de novo DNA and protein synthesis in infected epithelial cells, suggest that *S. flexneri* induces an arrest in the host cell cycle.

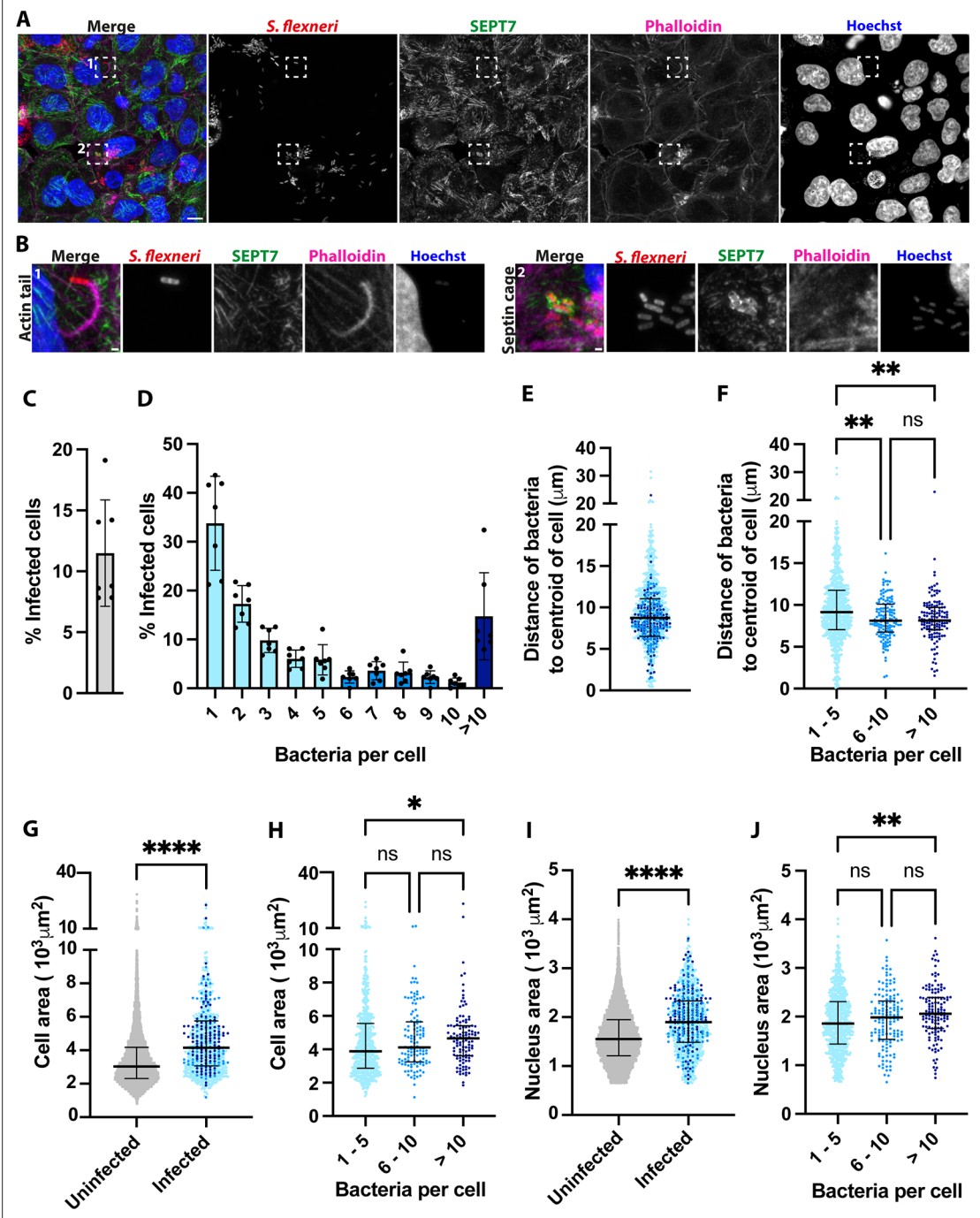

**Figure 1.** High-content imaging captures the heterogeneity of *S. flexneri* infection. (**A**) Representative high-resolution microscopy image of HeLa cells infected with *S. flexneri* expressing mCherry acquired with the high-content microscope ZEISS CellDiscoverer 7. Scale bar, 10 µm. (**B**) Enlarged view of insets in A, highlighting a *S. flexneri* bacteria polymerising an actin tail (1) and entrapped in a septin cage (2), scale bar, 1 µm. (**C**) Percentage of infected cells at 3 hpi of 7 independent experiments. Graph represents mean = 11.49% ± SD=4.36%. (**D**) Percentage of infected cells harbouring different bacterial doses, graph represents mean ± SD. (**E**) Distance of the bacteria to the centroid of the host cell. Graph represents the median (8.74 µm) and interquartile range (6.51–11.05 µm). Values from n=4990 cells from 7 independent experiments. (**F**) Distance of individual bacteria to the centroid of the host cell depending on its bacterial load. Graph represents the median and interquartile range. Kruskal-Wallis test and Dunn's multiple comparisons test. (**G**) Area of cells depending on infection. Graph represents the median area (uninfected: 3031 µm², infected: 4148 µm²) and interquartile range (uninfected: 2308–4169 µm², infected: 3062–5752 µm²). Values from n=9469 uninfected cells and n=1031 infected cells from 7 independent experiments. (**H**) Area of infected cells depending on their bacterial load. Graph represents the median and interquartile range. Kruskal-Wallis test and Dunn's multiple comparisons test. (**I**) Nuclear area of cells depending on infection. Graph represents the median area (uninfected: 1551 µm², infected: 1897

*Figure 1 continued on next page*

Figure 1 continued

µm²) and interquartile range (uninfected: 1209–1950 µm², infected: 1486–2333 µm²). Values from n=9469 uninfected cells and n=1031 infected cells from 7 independent experiments. (**J**) Nuclear area of infected cells depending on its bacterial load. Graph represents the median and interquartile range. Kruskal-Wallis test and Dunn's multiple comparisons test.

The online version of this article includes the following figure supplement(s) for figure 1:

**Figure supplement 1.** S.*flexneri* infection induces host cell size changes.

## Design of a deep learning pipeline to automatically identify bacteria interacting with SEPT7

*S. flexneri* has been shown to interact with septins during infection. These interactions include septin ring-like structures surrounding phagocytic cups, autophagosomes, actin tails, and protrusions, all being morphologically diverse structures that are normally scored by hand (*Krokowski et al., 2018*; *Lobato-Márquez et al., 2023*; *Mostowy et al., 2010*). To study septin-associated bacteria using high-content microscopy, we designed a tailored analysis pipeline for the automatic and unbiased identification of SEPT7-bacteria interactions (*Figure 3*). SEPT7 is a core component of septin hexamers and octamers and is historically used to identify septin structures (*Mostowy et al., 2010*). The analysis pipeline includes two deep learning models based on convolutional neural networks (CNNs), which are effective in solving classification tasks in computer vision (*Krizhevsky et al., 2012*; *LeCun et al., 2015*; *Nielsen, 2015*). As septins can surround individual and also multiple bacteria simultaneously (*Figure 4—figure supplement 1A*), we focused on individual bacteria. For this, we trained the first classification model to discriminate between 'single' bacteria and 'clumps'. The second classification model was designed to assign those single bacteria into 'SEPT7-positive structures' or alternatively 'negatives', according to the morphology of SEPT7 recruitment around them.

The first classification model (discerning 'single' vs 'clumped' bacteria) together with its training and performance metrics is summarised in *Figure 4*. A large dataset of annotated bacteria (*Figure 4A and B*, *Figure 4—figure supplement 1B*) was used to train a sequential model with a typical CNN architecture that was experimentally tested and optimised accordingly for best results (*Figure 4C*). This model provided a good fit of the training data (high accuracy, low loss), with similar predictive power for the validation dataset (*Figure 4D and E*). Performance of the model was further assessed with a confusion matrix where a separate annotated test dataset was challenged and correct number of predictions scored, showing that 96% of SEPT7 structures and 83% of clumps were correctly identified (*Figure 4F*). Finally, effective performance of the model was ensured with high values achieved in additional metrics testing for type I and II errors (precision, recall, F1 score) (*Figure 4G*).

The second classification task (identification of SEPT7 'positive' vs 'negative' bacteria) was more challenging due to their intrinsic heterogeneity plus the presence of ubiquitous SEPT7 filaments that populate the vicinity of bacteria (but do not associate with bacteria per se). To address these challenges, we increased the amount of data and complexity of the second classification model (*Figure 4H–J*, *Figure 4—figure supplement 1C*) compared to the first (*Figure 4A–C*). This included more than 1200 images of *S. flexneri* associated with SEPT7, both from infections with wild-type bacteria or expressing Afal to increase bacterial invasion. The training process achieved a high accuracy and low loss for the training and validation dataset (*Figure 4K and L*). Correct prediction of manually annotated SEPT7 positive structures was 80% as shown in the confusion matrix (*Figure 4M*), with performance metrics precision, recall, and F1 score also around 0.8 (*Figure 4N*).

## Characterisation of the morphological heterogeneity of SEPT7 recruitment to *S. flexneri*

The training of the classification model to identify bacteria associated with SEPT7 required a large dataset of images. We observed diverse morphological patterns of SEPT7 signal around bacteria (*Figure 5A–C*). We differentiated five frequent subcategories from a total of 855 cases. The first corresponds to most cases (66.7%), where SEPT7 forms rings that entrap bacteria transversally (*Figure 5A and B, a*). We also observed two categories where SEPT7 surrounds bacteria more homogeneously either in a tight (18.2%) or loose association (6.1%) (*Figure 5A and B, b–c*). Finally, we observed partial SEPT7 recruitment, with only one (5.0%) or two (4.0%) bacterial poles being targeted (*Figure 5A and B, d–e*). Given this diversity, we combined the entire dataset to summarise the probability of SEPT7

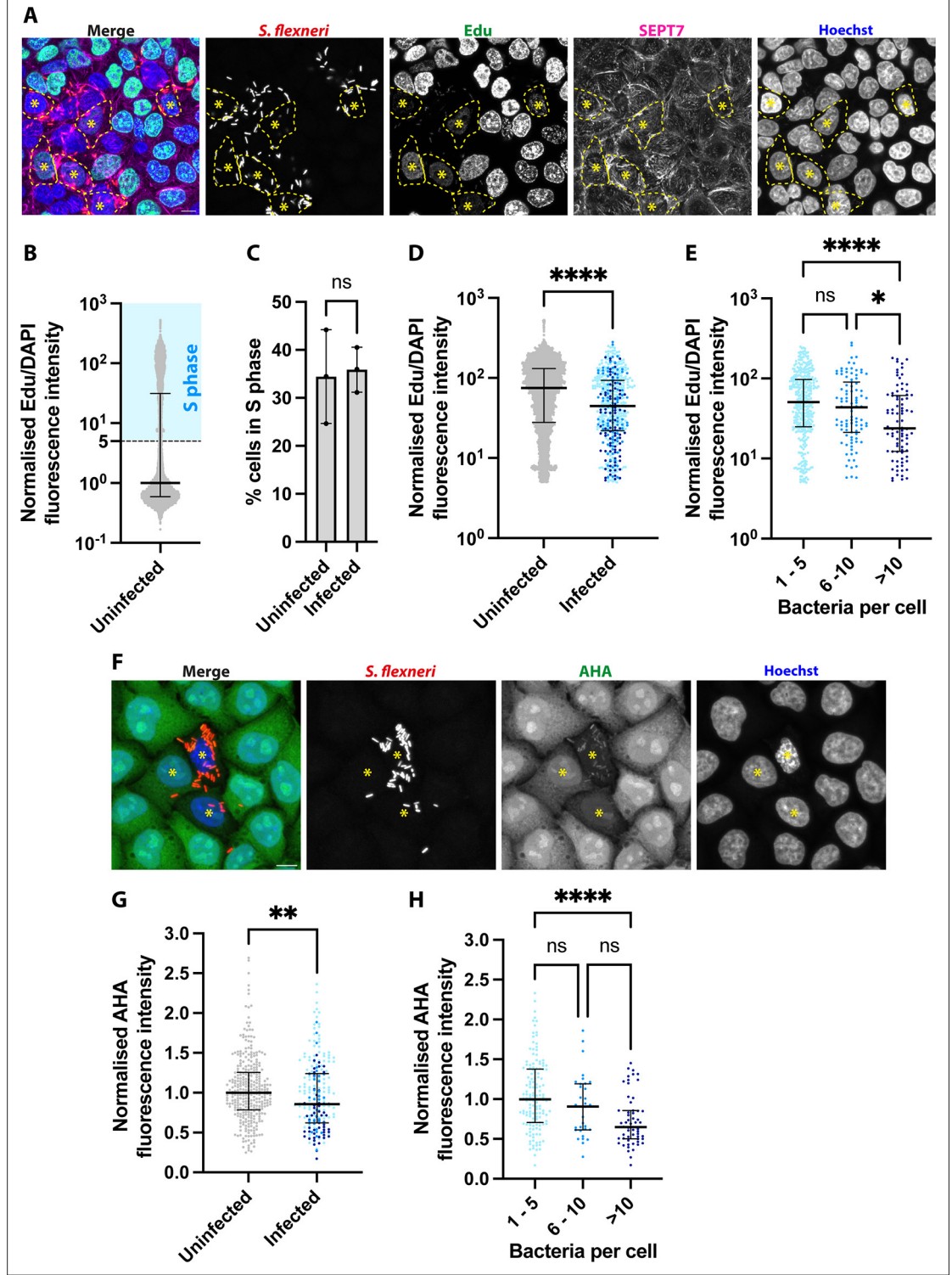

**Figure 2.** *S. flexneri* infection reduces host DNA synthesis and host protein synthesis. (**A**) Representative microscopy image of HeLa cells infected with *S. flexneri* expressing mCherry at 3 hpi after 1 hr incubation with Click-iT EdU. Asterisks label the nucleus of infected cells with reduced EdU incorporation. Contour of selected cells is delimited with a dashed line. Scale bar, 10 μm. (**B**) EdU acquisition in uninfected cells is bimodal. Graph represents the normalised median of the ratio between the fluorescence intensity of EdU and Hoechst (1), and interquartile range (0.5924–31.22). Values from n=8941 cells from 3 independent experiments. An arbitrary threshold of 5 is selected to define cells in S-phase. (**C**) Frequency of cells in S-phase (as defined in B) depending on infection. Graph represents mean ± SD (uninfected: 34.45% ± 9.78%, infected: 35.89% ± 4.69%). Student's t-test. (**D**) EdU acquisition in cells in S-phase depending on infection. Graph represents the normalised median of the ratio between the fluorescence intensity of EdU

*Figure 2 continued on next page*

*Figure 2 continued*

and Hoechst (uninfected: 75.14, infected: 44.73), and interquartile range (uninfected: 27.89–131.0, infected: 22.05–93.36), from 3065 uninfected and 552 infected cells. Mann-Whitney U test. (**E**) EdU acquisition in cells in S-phase depending on bacterial load. Graph represents the median and interquartile range. Kruskal-Wallis test and Dunn's multiple comparisons test. (**F**) Representative microscopy image of HeLa cells infected with *S. flexneri* expressing mCherry at 3 hpi after 1 hr incubation with Click-iT AHA. Asterisks label the infected cells with reduced AHA incorporation. Scale bar, 10 μm. (**G**). AHA acquisition in host cells depending on infection. Graph represents the normalised median of AHA fluorescence intensity (uninfected: 1, infected: 0.86), and interquartile range (uninfected: 0.7850–2.694, infected: 0.6207–1.241), from 36 uninfected and 233 infected cells. Mann-Whitney U test. (**H**) AHA acquisition in infected cells depending on bacterial load. Graph represents the median and interquartile range. Kruskal-Wallis test and Dunn's multiple comparisons test.

The online version of this article includes the following figure supplement(s) for figure 2:

**Figure supplement 1.** Click-iT EdU and Click-iT AHA specifically label de novo DNA and protein synthesis in HeLa cells.

interactions with bacteria (*Figure 5—figure supplement 1*). As bacteria are found in any orientation inside epithelial cells, we included only those instances where the z-projection of the bacterial axis measured more than 2.5 μm, ensuring they were positioned horizontally when imaged. Representations shown in *Figure 5—figure supplement 1A and C* and graphs in *Figure 5—figure supplement 1B and D* can be understood as a probability map of the area of influence of SEPT7 when associating with intracellular *S. flexneri*.

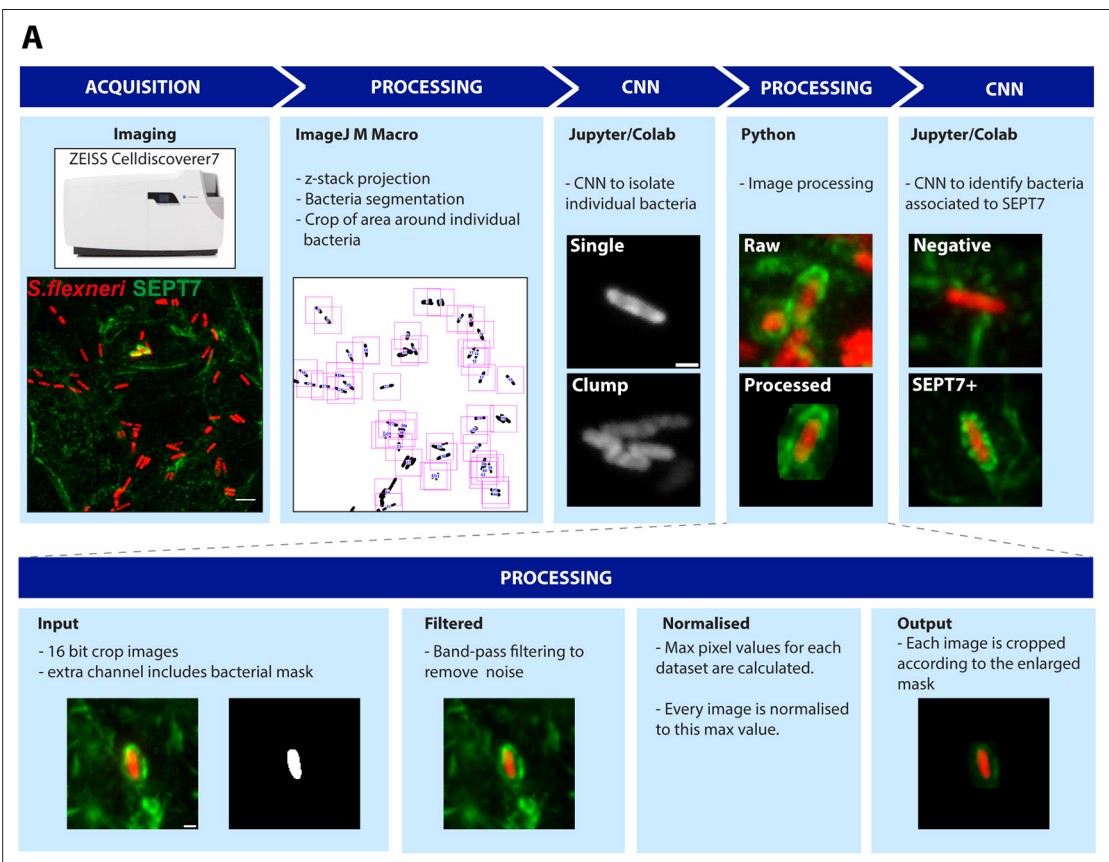

**Figure 3.** Microscopy pipeline to automatically identify SEPT7-*S. flexneri* interactions. (**A**) Diagram representing the steps involved in the imaging and analysis pipeline. After infection, fixation, and staining, a high number of images are automatically acquired using the high-content microscope. Scale bar, 10 μm. These are then processed to segment bacteria based on their fluorescence, so that a square field containing the bacteria in the centre is cropped and saved for subsequent analysis. A deep learning model based on convolutional neural network (CNN) is applied using Jupyter notebooks and the Python Keras library hosted on Colab to sort individual, isolated bacteria. Several processing steps are applied using Python to remove noise (band-pass filtering by difference of Gaussian and mean filters), normalise the data across datasets for comparison, and remove signal away from the bacteria which is irrelevant for the identification of SEPT7 assemblies. Scale bar, 1 μm. Finally, a similar second deep learning model based on CNN is applied to identify SEPT7 interacting with bacteria.

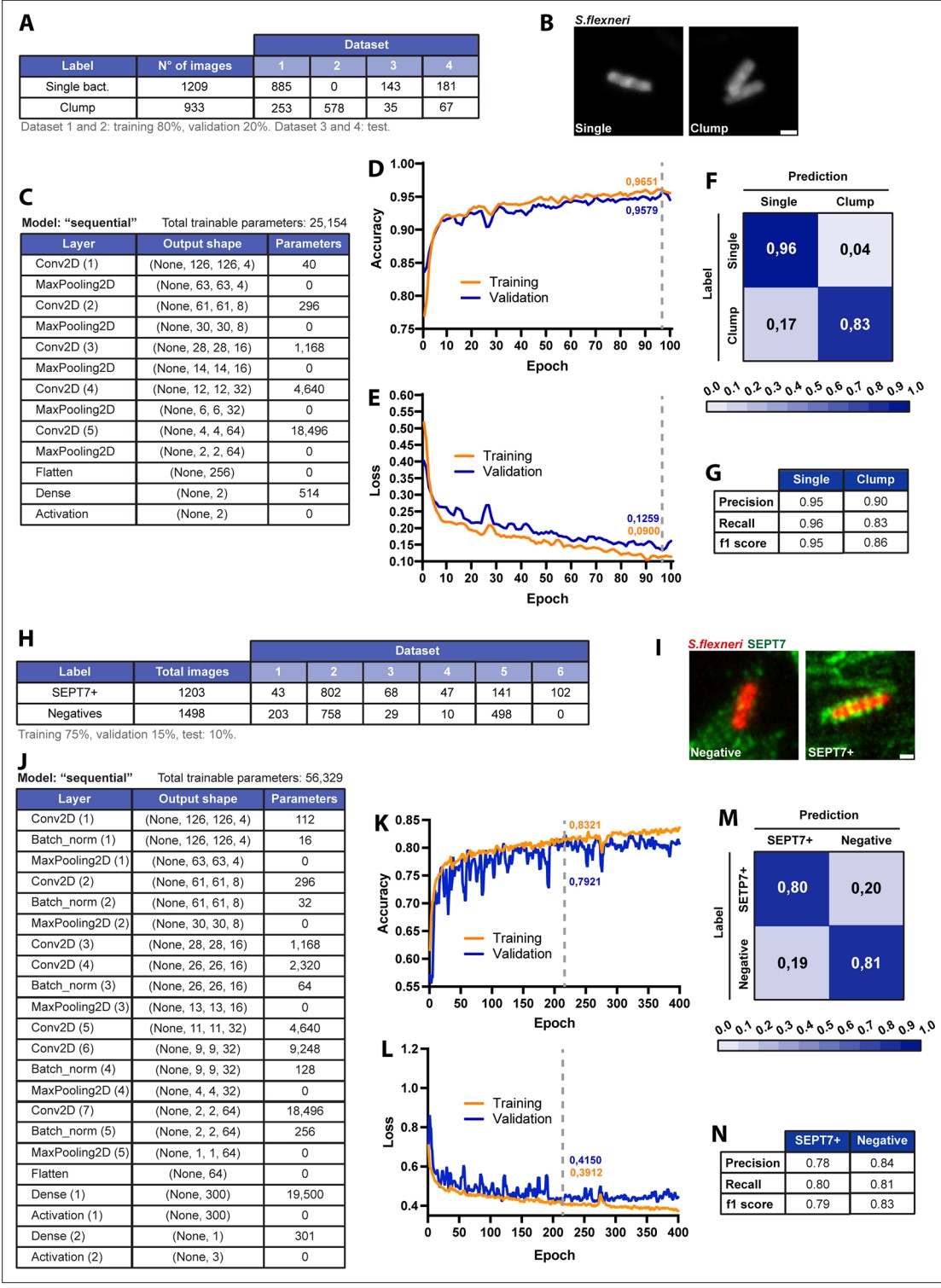

**Figure 4.** Training of a deep learning algorithm to identify single isolated bacteria cells and SEPT7-associated *S. flexneri*. (**A**) Table summarising the annotated dataset used for the training process. The labels of the two classification classes used were 'Single bacteria' and 'Clump'. The annotated images were randomly split into two groups, one used as training dataset (comprising 80% of the images) and another used as validation dataset (with the remaining 20%). A separate annotated dataset was used for testing. (**B**) Representative Airyscan images of bacteria used for each class. 2D max projections of z-stacks were used for training. Scale bar, 1 µm. (**C**) Architecture of the deep learning algorithm used for training. Each row describes the characteristics of the sequential transformation steps applied. Conv2D stands for 2D Convolution. (**D, E**) Accuracy and Loss as metrics to represent the training process over subsequent epochs (entire passing of the training data through the algorithm). Accuracy increases and loss decreases for both

*Figure 4 continued on next page*

*Figure 4 continued*

training and validation datasets, indicating a good fit of the model to the data. Vertical dashed grey line indicates Early Stopping, or epoch value for the minimum validation Loss. (**F**) Confusion matrix performed on an annotated test dataset not used previously for the training or validation process, indicating the percentage of predictions that were correct or wrong for each class. (**G**) Precision, recall, and F1 score as metrics that summarise the performance of the model or classifier. (**H**) Table summarising the annotated dataset used for the training process. The labels of the two classification classes used were 'Septin' and 'Negative'. The annotated images were randomly split into three groups, one used as training dataset (comprising 75% of the images), another used as validation dataset (with 15%), and a last one used as test dataset (with 10%). Due to the data being very imbalanced (15% natural frequency of SEPT7-associated bacteria), the images comprised in the Negative class were under-sampled as indicated in the table. (**I**) Representative Airyscan images of bacteria used for each class. 2D max projections of z-stacks were used for training. Scale bar, 1 µm. (**J**) Architecture of the deep learning algorithm used for training. Each row describes the characteristics of the sequential transformation steps applied. Conv2D stands for 2D Convolution, Batch_norm stands for Batch normalisation. (**K, L**) Accuracy and Loss as metrics to represent the training process over subsequent epochs. Accuracy increases and loss decreases for both training and validation datasets, indicating a good fit of the model to the data. Vertical dashed grey line indicates Early Stopping or epoch value for the minimum validation Loss. (**M**) Confusion matrix performed on the test dataset not used previously for the training or validation process, indicating the percentage of predictions that were correct or wrong for each class. (**N**) Precision, recall, and F1 score as metrics that summarise the performance of the model or classifier.

The online version of this article includes the following figure supplement(s) for figure 4:

**Figure supplement 1.** Additional examples of the annotated datasets for deep learning.

## SEPT7 is recruited to *S. flexneri* with increased T3SS activity

Septins have previously been shown to recognise growing bacterial cells in vitro and during infection of HeLa cells, using pharmacologic and genetic manipulation (*Krokowski et al., 2018*; *Lobato-Márquez et al., 2021*). To test the metabolic state of SEPT7-associated *S. flexneri* at 3 hr 40 min post infection (hpi), we used our high-content imaging and analysis pipeline (*Figure 6*). We measured *S.*

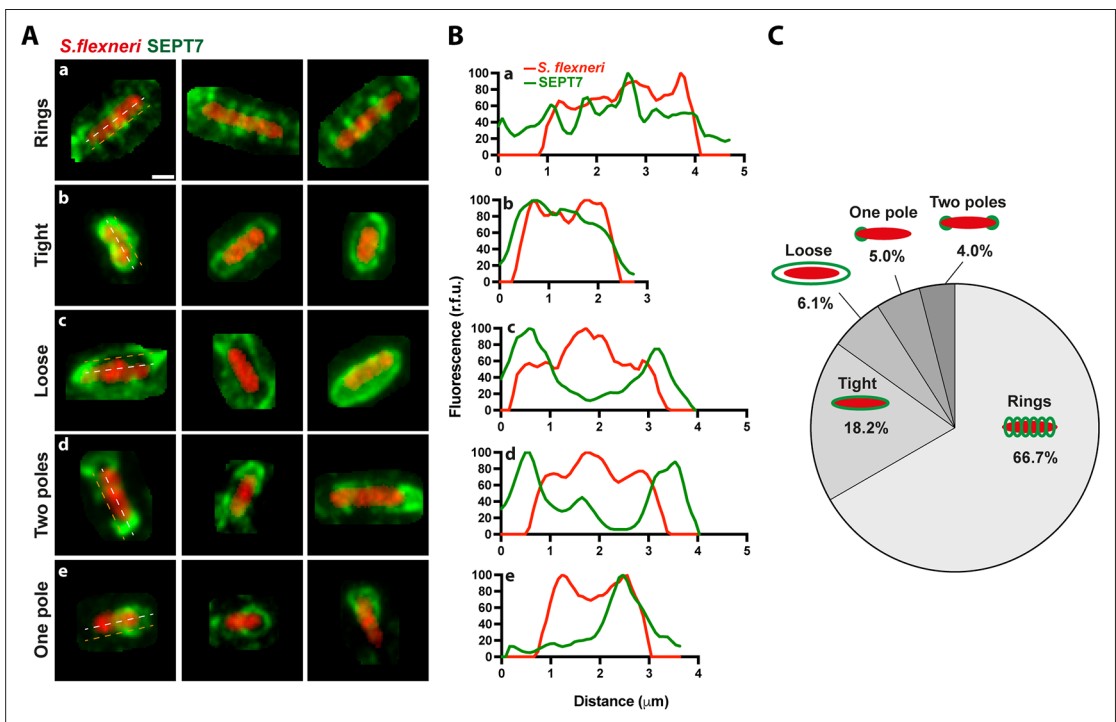

**Figure 5.** Septin assemblies associated with *S. flexneri* are heterogeneous. (**A**) Selected examples of the training dataset to showcase the diverse morphology of septin assemblies associated with bacteria. Frequent morphologies include (**a**) ring-like structures around the bacteria, (**b**) smooth tight recruitment around the entire bacterial surface, (**c**) loose structures more distant from the bacterial surface, (**d**) septin recruitment to both bacterial poles, (**e**) septin recruitment to a single bacterial pole. Scale bar, 1 µm. (**B**) Intensity profiles of the cytosolic bacterial mCherry and the SEPT7 recruitment, as indicated in the white and orange dashed lines, respectively. (**C**) Pie chart representing the relative frequency of 855 *S. flexneri*-SEPT7 associations depicted in A.

The online version of this article includes the following figure supplement(s) for figure 5:

**Figure supplement 1.** Probability maps of SEPT7 around *S. flexneri*.

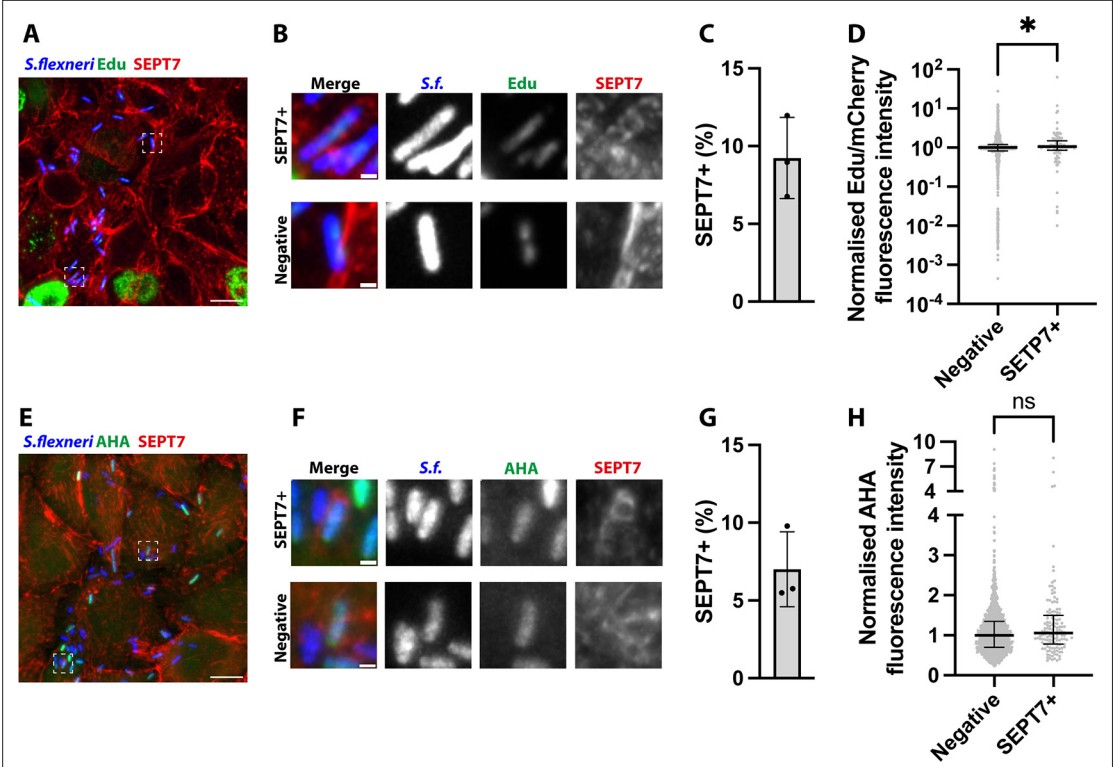

**Figure 6.** *S. flexneri* associated with SEPT7 is metabolically active. (**A**) Representative microscopy image of HeLa cells infected with *S. flexneri* expressing mCherry at 3 hpi after 1 hr incubation with Click-iT EdU. Scale bar, 10 µm. (**B**) Enlarged view of insets in A, highlighting a *S. flexneri* associated or not to SEPT7 structures. Scale bar, 1 µm. (**C**) Frequency of the number of SEPT7-associated bacteria identified using deep learning from experiments in A, from n=1024 from 3 independent replicates. Graph represents mean ± SD (9.23% ± 2.62%). (**D**) Normalised ratio between EdU signal and mCherry in individual bacteria. Graph represents the median (negatives: 1, septin associated: 1.059), and interquartile range (negatives: 0.8185–1.207, septin associated: 0.8573–1.487). Mann-Whitney U test. (**E**) Representative microscopy image of HeLa cells infected with *S. flexneri* expressing mCherry at 3 hpi after 1 hr incubation with Click-iT AHA. Scale bar, 10 µm. (**F**) Enlarged view of insets in E, highlighting a *S. flexneri* associated or not to SEPT7 assemblies. Scale bar, 1 µm. (**G**) Frequency of the number of SEPT7-associated bacteria identified using deep learning from experiments in E, from n=1872 bacteria from 3 independent replicates. Graph represents mean ± SD (7.01% ± 2.41%). (**D**) Normalised AHA signal in individual bacteria in A, B. Graph represents the median (negatives: 1, SEPT7 associated: 1.055) and interquartile range (negatives: 0.6986–1.345, SEPT7 associated: 0.7837–1.497). Mann-Whitney U test.

The online version of this article includes the following figure supplement(s) for figure 6:

**Figure supplement 1.** Click-iT EdU and Click-iT AHA specifically label de novo DNA and protein translation in *S. flexneri*.

*flexneri* de novo synthesis of DNA and proteins using a 1 hr pulse Click-iT EdU and Click-iT AHA. Only active bacteria incorporated these analogues, as the antibiotic nalidixic acid (inhibitor of the bacterial DNA gyrase; *Sugino et al., 1977*) and rifampicin (inhibitor of the bacterial RNA polymerase; *Ezekiel and Hutchins, 1968*) significantly reduced their incorporation, respectively (*Figure 6—figure supplement 1A–D*, *Figure 6—figure supplement 1E and F*). During infection, bacteria associated with SEPT7 showed heterogeneous levels of DNA synthesis and protein translation similar to SEPT7-negative bacteria (*Figure 6A–H*).

*S. flexneri* pathogenicity relies on its T3SS. To understand effector secretion at the individual bacterial level, we used *S. flexneri* expressing the transcription-based secretion activity reporter (TSAR) (*Campbell-Valois et al., 2014*). This reporter comprises a short-lived fast-maturing superfolder GFP (GFPmsf2) that is expressed in conditions where the T3SS is active, such as in broth upon addition of congo red (*Figure 7—figure supplement 1A and B*; *Parsot et al., 1995*) and during infection (*Figure 7*). We observed heterogeneity in TSAR signal (*Figure 7A and B*). T3SS activation was higher in bacteria found in cells with a low bacterial burden (less than 10 intracellular bacteria) (*Figure 7A and C*). This suggests the highest level of activation occurs in bacterial pioneers that colonise neighbouring cells, in agreement with previous reports (*Campbell-Valois et al., 2014*). We then assessed T3SS

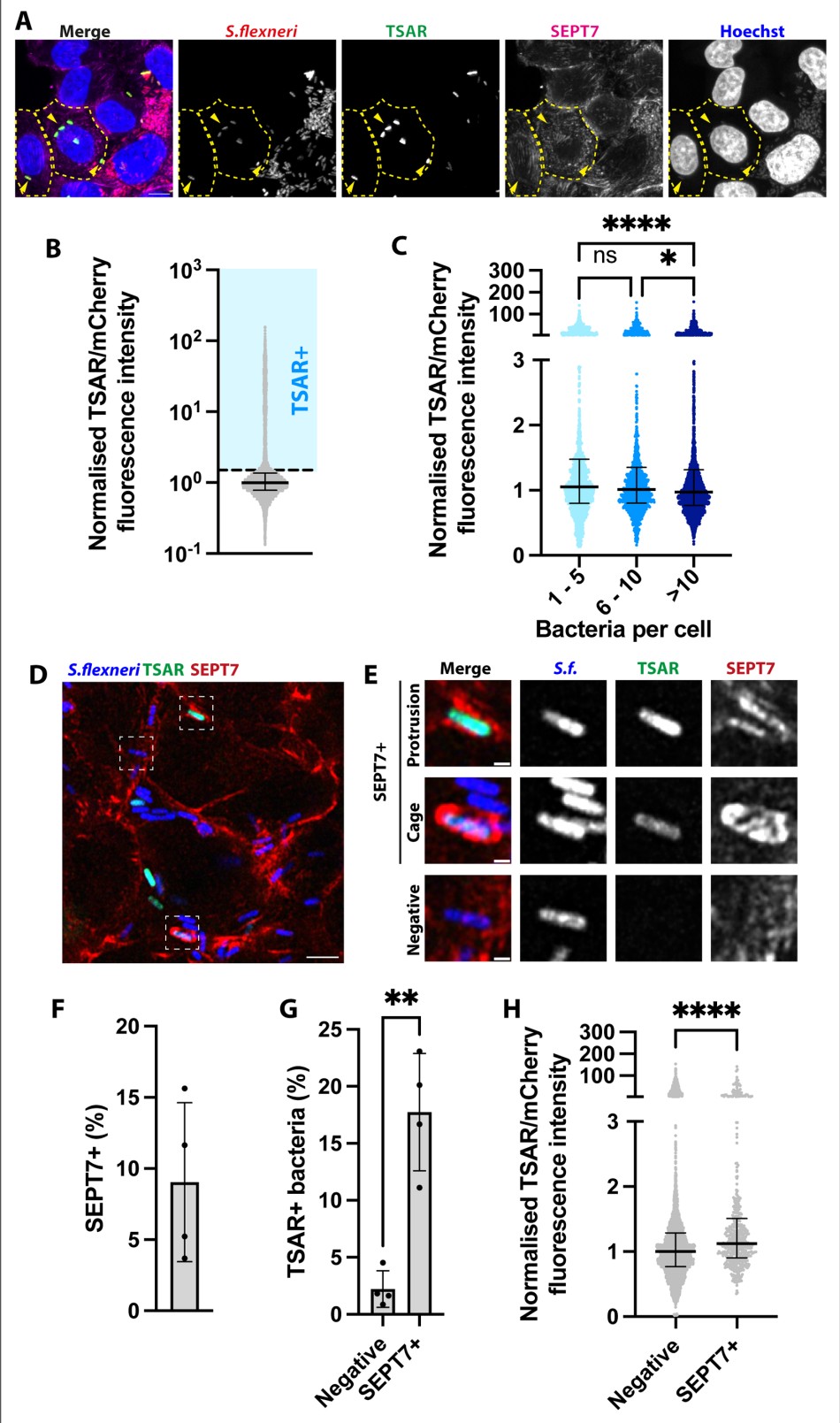

**Figure 7.** *S. flexneri* associated with SEPT7 has an active type III secretion system (T3SS). (**A**) Representative microscopy image of HeLa cells infected with *S. flexneri* expressing the transcription-based secretion activity reporter (TSAR) reporter at 3 hpi. Scale bar, 10 μm. Arroheads highlight TSAR+ bacteria. Selected infected cells' contours are delimited with dashed lines. (**B**) Distribution of TSAR signal intensity in individual bacteria. Graph

*Figure 7 continued on next page*

*Figure 7 continued*

represents the normalised median of the ratio between TSAR and mCherry (1), and interquartile range (0.7827–1.369). Values from n=3654 bacterial cells from 3 independent experiments. An arbitrary threshold of 1.5 is selected to define TSAR+ *S. flexneri*. (**C**) Intensity of TSAR signal in individual bacteria depending on infection load of host cells. Graph represents the normalised median and interquartile range. Kruskal-Wallis test and Dunn's multiple comparisons test. (**D**) Representative microscopy image of HeLa cells infected with *S. flexneri* expressing the TSAR reporter at 3 hpi. Scale bar, 10 μm. (**E**) Enlarged view of insets in D, highlighting a *S. flexneri* associated or not to SEPT7 assemblies. Scale bar, 1 μm. (**F**) Frequency of the number of SEPT7-associated bacteria identified using deep learning from experiments in F, from n=5030 bacteria from 4 independent replicates. Graph represents mean ± SD (9.044% ± 5.582%). (**G**) Percentage of TSAR+ *S. flexneri* depending on their association to SEPT7. Graph represents mean ± SD (negatives: 2.214±1.603, septin associated: 17.74±5.138). Student's t-test. (**H**) Normalised ratio between TSAR signal and mCherry in individual bacteria. Graph represents the median (negatives: 1, SEPT7 associated: 1.122) and interquartile range (negatives: 0.7674–1.286, SEPT7 associated: 0.9047–1.511). Mann-Whitney U test.

The online version of this article includes the following figure supplement(s) for figure 7:

**Figure supplement 1.** Transcription-based secretion activity reporter (TSAR) labels type III secretion system (T3SS) activity specifically.

---

activation in *S. flexneri* associated with SEPT7 using our deep learning analysis pipeline (*Figure 7D–H*). We detected a significant increase in TSAR-positive bacteria in SEPT7-associated *S. flexneri* compared to SEPT7-negative bacteria (*Figure 7G and H*). These include SEPT7-associated protrusions and cage-like structures (*Figure 7E*, *Figure 7—figure supplement 1*). These data underscore the antimicrobial role of septins in associating with actively secreting bacteria.

## Discussion

Unravelling the intrinsic heterogeneity of host and bacterial cells underlying the infection process is paramount to designing effective therapeutic strategies. High-content and high-throughput image analysis are ideal tools to capture this heterogeneity when coupled to automatic phenotypic image analysis (*Eliceiri et al., 2012*; *Mattiazzi Usaj et al., 2016*; *Pegoraro and Misteli, 2017*). Here, we use high-content high-resolution microscopy and automated image analysis to quantitatively dissect the heterogeneity of *S. flexneri* in epithelial cells at the single-cell and population level at unprecedented subcellular detail.

We discover that *S. flexneri* induces important morphological and physiological changes in epithelial cells after 3 hr 40 min post infection. These include a marked decrease of DNA synthesis in cells in S-phase. Consistent with the ability of bacteria to manipulate the host cell cycle, *S. flexneri* has been shown to induce genotoxic stress (*Bergounioux et al., 2012*) and block the G2/M phase transition (*Iwai et al., 2007*; *Wang et al., 2019*). We also observed a collective shutdown of protein expression, which happened in a bacterial burden-dependent manner. A recent report has shown that the OspC family of *S. flexneri* effectors ADP-riboxanates the host translation initiation factor 3, leading to translational arrest and the formation of stress granules (*Zhang et al., 2024*). Collectively, these observations suggest that *S. flexneri* infection induces host cell stress and arrests the progression of the cell cycle. In the natural *S. flexneri* niche of the human gut, such an arrest would prevent maturation and shedding of infected epithelial cells, thereby promoting efficient gut colonisation (*Iwai et al., 2007*).

To understand how bacteria are targeted by cell-autonomous immune defences, such as septin cage entrapment, it is necessary to investigate phenotypic characteristics of targeted bacteria and the heterogeneous recruitment of cell-autonomous immune factors. While traditional image segmentation tools fail to enable this task, rapidly evolving developments in the field of computer vision are democratising the use of deep learning and CNNs for multiple applications (*Berg et al., 2019*; *Hung et al., 2020*; *Stringer et al., 2021*; *von Chamier et al., 2021*; *Weigert et al., 2020*), including infection biology (*Davidson et al., 2021*; *Fisch et al., 2021*; *Touquet et al., 2018*). However, the analysis of complex datasets still requires a tailored approach, as shown here for bacterial interactions with the septin cytoskeleton (which is ubiquitous in the cell and forms diverse assemblies whose precise identification requires high resolution). We successfully implement a CNN-based approach for the automated classification of bacteria associated with SEPT7 assemblies using z-stacks projections, enabling the analysis of thousands of bacterial cells while saving time and preventing human bias.

Building upon this work, future studies will improve use of 3D data for CNN re-training instead of z-stack projections, to allow more precise identification of *S. flexneri*-septin interactions, particularly when bacteria are placed perpendicular to the imaging plane.

A large dataset of images depicting *S. flexneri*-SEPT7 interactions was acquired to train the deep learning algorithm and to characterise the diversity of these structures. We observe that the most frequent septin assembly is in the form of rings transversally surrounding the bacterial surface. We also detected alternative morphologies, including homogeneous SEPT7 labelling surrounding the bacterial surface (tight vs loose), as well as the discrete recruitment of SEPT7 to one or two bacterial poles. While previous work has shown that septin cage assembly around *S. flexneri* can start with the discrete recruitment of septins to one or two bacterial poles (*Krokowski et al., 2018*), it is not yet known if the different septin assemblies we describe here correspond to different stages or fates of the same structure or whether it marks separate biological events. To fully address this question, follow-up studies should include time-lapse analysis of septin cage formation. We use deep learning analysis to interrogate intracellular bacteria and show that *S. flexneri* associated with SEPT7 presents heterogeneous DNA and protein synthesis. In addition, we show that bacteria associated with SEPT7 have increased T3SS activity (as compared to bacteria not associated with SEPT7). T3SS activity has been shown to be higher in motile bacteria (*Campbell-Valois et al., 2014*), which reinforces the antimicrobial role of septins in counteracting actin tail motility. Altogether, our deep learning tool to analyse septin-bacteria interactions opens the door to future studies on interactions between bacteria and components of cell-autonomous immunity.

## Concluding remarks

Heterogeneity in a controlled lab environment is only a small representation of the broader heterogeneity underlying the clinical infection process, considering the genetic background of the patient, interactions with microbiota, etc. In the future, our pipeline can be applied to investigate septin-bacteria interactions among different *Shigella* spp. or clinically relevant strains, as well as other intracellular bacteria (e.g. *M. marinum, Staphylococcus aureus, Pseudomonas aeruginosa*). In addition, our pipeline can be adapted to identify and score other cytoskeletal structures (such as actin tails) or membrane organelles (such as autophagosomes) associated with intracellular bacteria. Ultimately, we envision that our automated microscopy workflows will help to screen pharmacologic or genetic libraries in a multiparametric manner and understand their impact on host and bacterial cells at the single-cell and population level. In this way, novel host and bacterial factors underlying heterogeneity and bacterial pathogenesis could be identified, leading to transformative therapeutic strategies to combat bacterial infection in humans.

# Materials and methods

**Key resources table**

| Reagent type (species) or resource | Designation | Source or reference | Identifiers | Additional information |
|---|---|---|---|---|
| Strain, strain background (*Shigella flexneri*) | *Shigella flexneri* srv. 5 a str. M90T | *Mostowy et al., 2010* | | |
| strain, strain background (*Shigella flexneri*) | *Shigella flexneri* srv. 5 a str. M90T mCherry | *Mostowy et al., 2013* | | Constitutively producing mCherry |
| Strain, strain background (*Shigella flexneri*) | *Shigella flexneri* srv. 5 a str. M90T afaI | *Mostowy et al., 2010* | | Constitutively producing the adhesin AfaE |
| Strain, strain background (*Shigella flexneri*=) | *Shigella flexneri* srv. 5 a str. M90T GFP | *Mostowy et al., 2010* | | Constitutively producing GFP |
| Strain, strain background (*Shigella flexneri*) | *Shigella flexneri* srv. 5 a str. M90T GFP afaI | This study | | Constitutively producing the adhesin AfaE and GFP |
| Cell line (*Homo sapiens*) | HeLa | ATCC | CCL2 | |
| Antibody | Anti-human SEPT7 (rabbit polyclonal) | IBL | Cat#: 18991 RRID:AB_10705434 | IF (1:100) |

*Continued on next page*

*Continued*

| Reagent type (species) or resource | Designation | Source or reference | Identifiers | Additional information |
|---|---|---|---|---|
| Recombinant DNA reagent | pTSAR1ud2.4s (plasmid) | *Campbell-Valois et al., 2014* | | T3SS reporter |
| Recombinant DNA reagent | pAC-P$_{lac}$ (plasmid) | *Lobato-Márquez et al., 2015* | | pACYC184 derived plasmid. |
| Recombinant DNA reagent | pAC-AfaI-AmpR (plasmid) | This study | | For constitutive expression of the adhesin AfaE, ori p15A, Amp$^R$ |
| Recombinant DNA reagent | pAC-AfaI-CmR (plasmid) | This study | | For constitutive expression of the adhesin AfaE, ori p15A, Cm$^R$ |
| Sequence-based reagent | afaI-operon-fw | This study | PCR-primer | tgcgttgcgcgaaga tcctttgatcttttctacggggg |
| Sequence-based reagent | afaI-operon-rv | This study | PCR-primer | ggaagctaaac gagcccgatcttcccca |
| Sequence-based reagent | p15A-ori-fw | This study | PCR-primer | atcgggctcgttt agcttccttagctcc |
| Sequence-based reagent | p15A-ori-rv | This study | PCR-primer | aaggatcttcgcg caacgcaattaatgtaag |
| Sequence-based reagent | afaI-p15A-fw | This study | PCR-primer | cggggcgtaactg tcagaccaagtttac |
| Sequence-based reagent | afaI-p15A-rv | This study | PCR-primer | ttttctccatactctt ccttttcaatattattg |
| Sequence-based reagent | CmR-fw | This study | PCR-primer | aaggaagagtatggag aaaaaatcactggata taccaccgttgatatatccc |
| Sequence-based reagent | CmR-rv | This study | PCR-primer | ggtctgacagttac gccccgccctgcca |
| Commercial assay or kit | Click-iT EdU Cell Proliferation Kit for Imaging | ThermoFisher Scientific | Cat#: C10337 | |
| Commercial assay or kit | Click-iT AHA Protein Synthesis HCS Assay | ThermoFisher Scientific | Cat#: C10289 | |
| Chemical compound, drug | Congo red | Sigma-Aldrich | Cat#:C6767 | 0.01% (w/v) |
| Chemical compound, drug | Aphidicolin | Sigma-Aldrich | Cat#: A0781 | 50 µg/mL |
| Chemical compound, drug | Cycloheximide | Cell Signalling | Cat#: 2112 | 50 µg/mL |
| Chemical compound, drug | Nalidixic acid | Sigma-Aldrich | Cat#: N8878 | 2 µg/mL |
| Chemical compound, drug | Rifampicin | Sigma-Aldrich | Cat#: R3501 | 100 µg/ml |
| Software, algorithm | CellProfiler | CellProfiler | | v.4.0.7 |
| Other | Alexa-488-conjugated phalloidin | ThermoFisher Scientific | Cat#: 10729174 | IF (1:500) |
| Other | Hoescht | ThermoFisher Scientific | Cat#: H3570 | IF (1:500) |

**Table 1.** Bacterial strains and plasmids used in this study.

| Bacterial strain | Genotype | Reference |
|---|---|---|
| *Shigella flexneri* srv. 5a str. M90T | SmR | *Mostowy et al., 2010* |
| *S. flexneri* mCherry | Constitutively producing mCherry, Carb[R], Sm[R] | *Mostowy et al., 2013* |
| *S. flexneri afaI* | Constitutively producing the adhesin AfaE, Amp[R], Sm[R] | *Mostowy et al., 2010* |
| *S. flexneri* GFP | Constitutively producing GFP, Carb[R] | *Mostowy et al., 2010* |
| *S. flexneri* GFP *afaI* | Constitutively producing the adhesin AfaE and GFP, Amp[R] Cm[R], Sm[R] | This study |
| **Plasmid** | **Characteristics** | **Reference** |
| pTSAR1ud2.4s | T3SS reporter | *Campbell-Valois et al., 2014* |
| pAC-P$_{lac}$ | pACYC184-derived plasmid | *Lobato-Márquez et al., 2015* |
| pAC-AfaI-AmpR | For constitutive expression of the adhesin AfaE, ori p15A, Amp[R] | This study |
| pAC-AfaI-CmR | For constitutive expression of the adhesin AfaE, ori p15A, Cm[R] | This study |
| **Primer** | **Sequence (5′–3′)** | **Reference** |
| *afaI-operon-fw* | tgcgttgcgcgaagatcctttgatcttttctacgggg | This study |
| *afaI-operon-rv* | ggaagctaaacgagcccgatcttcccca | This study |
| *p15A-ori-fw* | atcgggctcgtttagcttccttagctcc | This study |
| *p15A-ori-rv* | aaggatcttcgcgcaacgcaattaatgtaag | This study |
| *afaI-p15A-fw* | cggggcgtaactgtcagaccaagtttac | This study |
| *afaI-p15A-rv* | ttttctccatactcttccttttttcaatattattg | This study |
| *CmR-fw* | aaggaagagtatggagaaaaaaatcactggatataccaccgttgatatatccc | This study |
| *CmR-rv* | ggtctgacagttacgccccgccctgcca | This study |

## Reagents

The following antibodies were used: rabbit anti-SEPT7 (1:100 dilution, #18991, IBL), Alexa-488-conjugated anti-rabbit antibody (1:500, #10729174, Thermo Fisher Scientific), Alexa-647-conjugated anti-rabbit antibody (1:500, #A27040, Thermo Fisher Scientific). The following dyes were used: Alexa-647-conjugated phalloidin (1:500, #10656353, Thermo Fisher Scientific), Hoechst (1:500, #H3570, Thermo Fisher Scientific), 0.01% (wt/vol) congo red (#C6767, Sigma-Aldrich). The following drugs were used: 50 µg/mL aphidicolin (#A0781 Sigma-Aldrich), 50 µg/mL cycloheximide (#2112 Cell Signaling), 2 µg/mL nalidixic acid (#N8878 Sigma-Aldrich), and 100 µg/mL rifampicin (#R3501 Sigma-Aldrich).

## Bacterial strains and culture conditions

The bacterial strains and plasmids described in this study are listed in *Table 1*. *S. flexneri* 5a str. M90T mCherry was used throughout the manuscript unless indicated otherwise. *S. flexneri* strains were grown in trypticase soy broth (TCS) agar containing congo red to select for red colonies, indicative of a functional T3SS. Conical polypropylene tubes (#CLS430828, Corning) containing 5 mL of TCS were inoculated with individual red colonies of *S. flexneri* and were grown ~16 hr at 37°C with shaking at 200 rpm. The following day, bacterial cultures were diluted in fresh prewarmed TCS (1:50 or 1:100 vol/vol) and cultured until an optical density (OD, measured at 600 nm) of 0.6.

## Cloning

Primers used in this study were designed using NEBuilder Assembly Tool (https://nebuilder.neb.com/#!/). To produce *S. flexneri* expressing AfaI and GFP, the pACAfaI plasmid was constructed (plasmid characteristics and primer sequences are located in *Table 1*). Briefly, the *afaI* operon together with the *amp$^R$* gene was amplified from *S. flexneri afaI* strain using the primers *afaI-operon-fw* and *afaI-operon-rv*. The *ori p15A* was amplified from pAC-P$_{lac}$ using the primers *p15A-ori-fw* and *p15A-ori-fw*.

Both DNA fragments were assembled using the Gibson assembly to generate the plasmid pAC-AfaI-AmpR. In a second step, the *AmpR* gene from pAC-AfaI-Amp was substituted by the *CmR* gene and generated pAC-AfaI-Cm. This was achieved by amplification of the *CmR* gene from pNeae2 using the primers *CmR-fw CmR-rv* and amplification of the backbone of pAC-AfaI-Amp with primers *afaI-p15A-fw, afaI-p15A-rv*, followed by Gibson assembly. Gibson assembly was performed at 50°C for 30 min using the HiFi DNA Assembly Master Mix (#E2621L, New England Biolabs). Resulting plasmid pAC-AfaI-Cm was transformed into *S. flexneri* GFP by electroporation.

## Cell lines

HeLa (ATCC CCL-2) cells were grown at 37°C and 5% $CO_2$ in Dulbecco's modified Eagle medium (DMEM, Gibco) supplemented with 10% fetal bovine serum (FBS, Sigma-Aldrich). Cell lines were tested for mycoplasma infection and tested negative.

## Infection of human cells

HeLa cells were seeded in 96-well microplates (PhenoPlates, Perkin Elmer) at a confluency of $10^4$ cells/well 2 days prior to infection. Alternatively, $9×10^4$ HeLa cells were seeded in six-well plates (Thermo Scientific) containing 22×22 $mm^2$ glass coverslips 2 days before the infection. Bacterial cultures were grown as described, and cell cultures were infected with *S. flexneri* strains as described previously (*Mostowy et al., 2010*). Briefly, Hela cells were infected with *S. flexneri* by spin inoculation at 110×*g* for 10 min at an MOI (bacteria:cell) of 100:1. Then, microplates were placed at 37°C and 5% $CO_2$ for 30 min. Infected cultures were washed 3× with phosphate-buffered saline (PBS) pH 7.4 and incubated with fresh DMEM containing 10% FBS and 50 mg/mL gentamicin at 37°C and 5% $CO_2$ for 3 or 4 hr. In order to obtain more instances of *S. flexneri* associated with SEPT7 in *Figure 4H*, one of the datasets was obtained after infection with *S. flexneri* expressing the adhesin AfaI (*Labigne-Roussel and Falkow, 1988*) and GFP to increase host cell invasion at an MOI of 10 in the absence of spin inoculation.

## Click chemistry

To label de novo DNA and protein synthesis, Click-iT EdU Cell Proliferation Kit for Imaging (#C10337 Thermo Fisher Scientific) and Click-iT AHA Protein Synthesis HCS Assay (#C10289 Thermo Fisher Scientific) were used, respectively, according to the manufacturer's instructions. In the case of AHA, cells were cultured in glutamine, cystine, and methionine-free DMEM (#21013024 Thermo Fisher Scientific) for 1 hr prior to the treatment. 10 µM EdU and 25 µM AHA were supplemented to live bacterial or mammalian cell cultures for 1 hr. Cells were fixed for immunofluorescence as indicated below. Click azide/alkyne reactions were performed after permeabilisation and prior to antibody staining as indicated below, according to the manufacturer's instructions.

## Immunofluorescence and fluorescence microscopy

Bacteria, coverslips, or 96-well microplates containing adherent infected or uninfected cells were washed three times with PBS pH 7.4 and fixed for 15 min in 4% paraformaldehyde (in PBS) at RT. Fixed cells were washed three times with PBS pH 7.4 and subsequently permeabilised for 5 min with 0.1% Triton X-100 (in PBS). Cells were then washed three times in PBS and incubated 1 hr with primary antibodies diluted in PBS supplemented with 0.1% Triton X-100 and 1% bovine serum albumin. Cells were then washed three times in PBS and incubated for 45 min with anti-rabbit secondary antibodies diluted 0.1% Triton X-100 (in PBS), and Hoescht and Alexa-conjugated phalloidin where indicated. 96-Well microplates were washed three times in PBS and preserved in 0.01% sodium azide (#10338380 Thermo Scientific) in PBS pH 7.4. Stained bacteria cultures and coverslips were washed three times with PBS pH 7.4 mounted on glass slides with ProLong Diamond Antifade Mountant (#P36970I Invitrogen). Fluorescence microscopy on stained bacteria was performed using a 63×/1.4 C-Plan Apo oil immersion lens on a Zeiss LSM 880 confocal microscope driven by ZEN Black software (v2.3). Microscopy images were obtained using z-stack image series taking 8–16 slices. Fluorescence microscopy on infected or uninfected cells was performed using a ZEISS Plan-APOCHROMAT 20×/0.95 Autocorr Objective or a ZEISS Plan-APOCHROMAT 50×/1.2 water immersion lens coupled to a 0.5x tubelens on a Zeiss CellDiscoverer 7 with Airyscan detectors driven by ZEN Blue software (v3.5). Microscopy

images were obtained using z-stack image series taking 32 slices. Confocal images were processed using Airyscan processing (Wiener filter) using 'Auto Filter' and '3D Processing' options.

## Flow cytometry

$5×10^4$ individual bacterial cells were analysed using flow cytometry with an LSRII flow cytometer (BD Biosciences). The data was analysed using FlowJo software, version 10.7.1. The median fluorescence of the bacterial population was determined and plotted for each staining.

## Analysis pipelines

Number of bacteria per cell, as well as fluorescence of bacteria, nuclei, and mammalian cells, was analysed using CellProfiler (v4.0.7) (*Carpenter et al., 2006*). Custom-made segmentation and deep learning pipelines for the classification of septin cages were written in ImageJ and Python using Keras.

## Statistics

All graphs were plotted, and statistical analysis was performed using Prism. n.s.: non-significant, *: p-value<0.05, **: p-value<0.01, ***: p-value<0.001.

## Code availability

Custom ImageJ macros, phyton scripts and Jupyter notebooks were annotated and deposited in Github (https://github.com/ATLopezJimenez/Toolset-high-content-analysis-of-Shigella-infection; copy archived at *Jiménez, 2024*).

# Acknowledgements

We thank members of the Mostowy and Brian Ho labs for helpful discussion. We thank François-Xavier Campbell-Valois for providing TSAR plasmids. We thank the instructors of the EMBL course Deep Learning for Image Analysis, in particular Prof. Pejman Rasti. ATLJ was funded by the Swiss National Science Foundation Early Postdoc.Mobility Fellowship (P2GEP3_188277) and the European Union's Horizon 2020 research and innovation programme under the Marie Skłodowska - Curie grant agreement no. H2020-MSCA-IF- 2020-895330. DB was supported by the Deutsche Forschungsgemeinschaft (DFG) Walter Benjamin Programme (BR 6637/1-1). GÖG was funded by the Human Frontier Science Program (LT000436/2021-L). This research in S Mostowy laboratory was supported by a Wellcome Trust Senior Research Fellowship (206444/Z/17/Z) and European Research Council Consolidator Grant (772853 - ENTRAPMENT).

# Additional information

### Funding

| Funder | Grant reference number | Author |
| --- | --- | --- |
| Swiss National Science Foundation | P2GEP3_188277 | Ana Teresa López-Jiménez |
| Horizon 2020 Framework Programme | H2020-MSCA-IF- 2020-895330 | Ana Teresa López-Jiménez |
| Deutsche Forschungsgemeinschaft | BR 6637/1-1 | Dominik Brokatzky |
| Human Frontier Science Program | LT000436/2021-L | Gizem Özbaykal Güler |
| Wellcome Trust | 10.35802/206444 | Serge Mostowy |
| European Research Council | 772853 - ENTRAPMENT | Serge Mostowy |

The funders had no role in study design, data collection and interpretation, or the decision to submit the work for publication. For the purpose of Open Access, the authors have applied a CC BY public copyright license to any Author Accepted Manuscript version arising from this submission.

## Author contributions
Ana Teresa López-Jiménez, Conceptualization, Data curation, Software, Formal analysis, Validation, Investigation, Visualization, Writing – original draft, Project administration, Writing – review and editing; Dominik Brokatzky, Conceptualization, Investigation, Methodology; Kamla Pillay, Tyrese Williams, Investigation; Gizem Özbaykal Güler, Software; Serge Mostowy, Conceptualization, Supervision, Funding acquisition, Writing – original draft, Writing – review and editing

## Author ORCIDs
Ana Teresa López-Jiménez ⓘ https://orcid.org/0000-0002-0289-738X
Dominik Brokatzky ⓘ https://orcid.org/0000-0001-6304-3292
Kamla Pillay ⓘ https://orcid.org/0009-0003-7519-1841
Gizem Özbaykal Güler ⓘ https://orcid.org/0000-0002-9115-7120
Serge Mostowy ⓘ https://orcid.org/0000-0002-7286-6503

Reviewer #2 (Public review): https://doi.org/10.7554/eLife.97495.3.sa1
Reviewer #4 (Public review): https://doi.org/10.7554/eLife.97495.3.sa2
Author response https://doi.org/10.7554/eLife.97495.3.sa3

# Additional files

## Supplementary files
MDAR checklist

## Data availability
This article is based on high-content microscopy data acquired with the ZEISS CellDiscoverer 7 automated microscope with Airyscan acquisition, which makes individual acquisition files extremely large. In addition, *S. flexneri* infects up to 15% of HeLa cells at the times observed and the frequency of entrapment in septin cages is around ~10%, so large datasets with hundreds of imaging fields were required. For example, to obtain ~1200 instances of *S. flexneri* entrapped in a septin cage (used to train the CNN in *Figure 4H*), we acquired >10,000 individual bacteria from a total of 6 different datasets, totalling around 10–15 TB just for this task. This scale makes public release of the complete raw set impractical. A representative dataset for each type of experiment performed and all the instances of bacteria that were used to train the CNN have been deposited in Dryad (https://doi.org/10.5061/dryad.6wwpzgn5z). Materials and additional raw microscopy data can be obtained from the corresponding authors.

The following dataset was generated:

| Author(s) | Year | Dataset title | Dataset URL | Database and Identifier |
|---|---|---|---|---|
| López-Jiménez A, Brokatzky D, Pillay K, Williams T, Özbaykal Güler G, Mostowy S | 2024 | High-content high-resolution microscopy and deep learning assisted analysis reveals host and bacterial heterogeneity during Shigella infection | https://doi.org/10.5061/dryad.6wwpzgn5z | Dryad Digital Repository, 10.5061/dryad.6wwpzgn5z |

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
