## [Editor Report · eLife Assessment]

This manuscript describes an AI-automated microscopy-based approach to characterize both bacterial and host cell responses associated with *Shigella* infection of epithelial cells. The methodology is **compelling** and should be helpful for investigators studying a variety of intracellular pathogens. The authors have acquired **important** findings regarding host and bacterial responses in the context of infection, which should be followed up with further mechanistic-based studies.

---

## [Referee Report · Reviewer #2 (Public review)]

Summary:

Septin caging has emerged as one of the innate immune response of eukaryotic cells to infections by intracellular bacteria. This fascinating assembly of eukaryotic proteins into complex structures restricts bacteria motility within the cytoplasm of host cells, thereby facilitating recognition by cytosolic sensors and components of the autophagy machinery. Given the different types of septin caging that have been described thus far, a single cell, unbiased approach to quantify and characterise septin recruitment at bacteria is important to fully grasp the role and function of caging. Thus, the authors have developed an automated image analysis pipeline allowing bacterial segmentation and classification of septin cages that will be very useful in the future, applied to study the role of host and bacterial factors, compare different bacterial strains or even compare infections by clinical isolates.

Strengths:

The authors developed a solid pipeline that has been thoroughly validated. When tested on infected cells, automated analysis corroborated previous observations and allowed the unbiased quantification of the different types of septin cages as well as the correlation between caging and bacterial metabolic activity. This approach will prove an essential asset in the further characterisation of septin cages for future studies.

Weaknesses:

As the main aim of the manuscript is to described the newly developed analysis pipeline, the results illustrated in the manuscript are essentially descriptive. The developed pipeline seems exceptionally efficient in recognising septin cages in infected cells but its application for a broader purpose or field of study remains limited.

---

## [Referee Report · Reviewer #4 (Public review)]

Summary

In this study, López-Jiménez and colleagues demonstrate the utility of using high-content microscopy in dissecting host and bacterial determinants that play a role in the establishment of infection using Shigella flexneri as a model. The manuscript nicely identifies that infection with Shigella results in a block to DNA replication and protein synthesis. At the same time, the host responds, in part, via the entrapment of Shigella in septin cages.

Strengths:

The main strength of this manuscript is its technical aspects. They nicely demonstrate how an automated microscopy pipeline coupled with artificial intelligence can be used to gain new insights regarding elements of bacterial pathogenesis, using Shigella flexneri as a model system. Using this pipeline enabled the investigators to enhance the field's general understanding regarding the role of septin cages in responding to invading Shigella. This platform should be of interest to those who study a variety of intracellular microbial pathogens.

Another strength of the manuscript is the demonstration - using cell biology-based approaches- that infection with Shigella blocks DNA replication and protein synthesis. These observations nicely dovetail with the prior findings of other groups. Nevertheless, their clever click-chemistry-based approaches provide visual evidence of these phenomena and should interest many.

Weaknesses:

There are two main weaknesses of this work. First, the studies are limited to findings obtained using a single immortalized cell line. It is appreciated that HeLa cells serve as an excellent model for studying aspects of Shigella pathogenesis and host responses. However, it would be nice to see that similar observations are observed with an epithelial cell line of intestinal, preferably colonic origin, and eventually, with a non-immortalized cell line, although it is appreciated that the latter studies are beyond the scope of this work.

The other weakness is that the studies are minimally mechanistic. For example, the investigators have data to suggest that infection with Shigella leads to an arrest in DNA replication and protein synthesis; however, no follow-up studies have been conducted to determine how these host cell processes are disabled. Interestingly, Zhang and colleagues recently identified that the Shigella OspC effectors target eukaryotic translation initiation factor 3 to block host cell translation (PMID: 38368608).

---

## [Author Response]

The following is the authors’ response to the original reviews

**Reviewer #1 (Public Review):**
Summary:In this study, López-Jiménez and colleagues demonstrated the utility of using high-content microscopy in dissecting host and bacterial determinants that play a role in the establishment of infection using Shigella flexneri as a model. The manuscript nicely identifies that infection with Shigella results in a block to DNA replication and protein synthesis. At the same time, the host responds, in part, via the entrapment of Shigella in septin cages.Strengths:The main strength of this manuscript is its technical aspects. They nicely demonstrate how an automated microscopy pipeline coupled with artificial intelligence can be used to gain new insights regarding elements of bacterial pathogenesis, using Shigella flexneri as a model system. Using this pipeline enabled the investigators to enhance the field's general understanding regarding the role of septin cages in responding to invading Shigella. This platform should be of interest to those who study a variety of intracellular microbial pathogens.Another strength of the manuscript is the demonstration - using cell biology-based approaches- that infection with Shigella blocks DNA replication and protein synthesis. These observations nicely dovetail with the prior findings of other groups. Nevertheless, their clever click-chemistry-based approaches provide visual evidence of these phenomena and should interest many.

We thank the Reviewer for their enthusiasm on technical aspects of this paper, regarding both the automated microscopy pipeline coupled with artificial intelligence and the click-chemistry based approaches to dissect DNA replication and protein synthesis by microscopy.

Weaknesses:There are two main weaknesses of this work. First, the studies are limited to findings obtained using a single immortalized cell line. It is appreciated that HeLa cells serve as an excellent model for studying aspects of Shigella pathogenesis and host responses. However, it would be nice to see that similar observations are observed with an epithelial cell line of intestinal, preferably colonic origin, and eventually, with a non-immortalized cell line, although it is appreciated that the latter studies are beyond the scope of this work.

The immortalized cell line HeLa is widely regarded as a paradigm to study infection by *Shigella* and other intracellular pathogens. However, we agree that future studies beyond the scope of this work should include other cell lines (eg. epithelial cells of colonic origin, macrophages, primary cells).

The other weakness is that the studies are minimally mechanistic. For example, the investigators have data to suggest that infection with Shigella leads to an arrest in DNA replication and protein synthesis; however, no follow-up studies have been conducted to determine how these host cell processes are disabled. Interestingly, Zhang and colleagues recently identified that the Shigella OspC effectors target eukaryotic translation initiation factor 3 to block host cell translation (PMID: 38368608). This paper should be discussed and cited in the discussion.

We appreciate the Reviewer’s concern about the lack of follow up work on observations of host DNA and protein synthesis arrest upon *Shigella* infection, which will be the focus of future studies. We acknowledge the recent work of Zhang et al. (Cell Reports, 2024) considering their similar results on protein translation arrest, and this reference has been more fully discussed in the revised version of the manuscript.

**Reviewer #2 (Public Review):**
Summary:Septin caging has emerged as one of the innate immune responses of eukaryotic cells to infections by intracellular bacteria. This fascinating assembly of eukaryotic proteins into complex structures restricts bacteria motility within the cytoplasm of host cells, thereby facilitating recognition by cytosolic sensors and components of the autophagy machinery. Given the different types of septin caging that have been described thus far, a single-cell, unbiased approach to quantify and characterise septin recruitment at bacteria is important to fully grasp the role and function of caging. Thus, the authors have developed an automated image analysis pipeline allowing bacterial segmentation and classification of septin cages that will be very useful in the future, applied to study the role of host and bacterial factors, compare different bacterial strains, or even compare infections by clinical isolates.Strengths:The authors developed a solid pipeline that has been thoroughly validated. When tested on infected cells, automated analysis corroborated previous observations and allowed the unbiased quantification of the different types of septin cages as well as the correlation between caging and bacterial metabolic activity. This approach will prove an essential asset in the further characterisation of septin cages for future studies.

We thank the Reviewer for their positive comments, and for highlighting the strength of our imaging and analysis pipeline to analyse *Shigella*-septin interactions.

Weaknesses:As the main aim of the manuscript is to describe the newly developed analysis pipeline, the results illustrated in the manuscript are essentially descriptive. The developed pipeline seems exceptionally efficient in recognising septin cages in infected cells but its application for a broader purpose or field of study remains limited.

The main objective of this manuscript is the development of imaging and analysis tools to study *Shigella* infection, and in particular, *Shigella* interactions with the septin cytoskeleton. In future work we will provide more mechanistic insight with novel experiments and broader applicability, using different cell lines (in agreement with Reviewer 1), mutants or clinical isolates of *Shigella* and different bacteria species (eg. *Listeria*, *Salmonella*, mycobacteria).

**Reviewer #3 (Public Review):**
Summary:The manuscript uses high-content imaging and advanced image-analysis tools to monitor the infection of epithelial cells by Shigella. They perform some analysis on the state of the cells (through measurements of DNA and protein synthesis), and then they focus on differential recruitment of Sept7 to the bacteria. They link this recruitment with the activity of the bacterial T3SS, which is a very interesting discovery. Overall, I found numerous exciting elements in this manuscript, and I have a couple of reservations. Please see below for more details on my reservations. Nevertheless, I think that these issues can be addressed by the authors, and doing so will help to make it a convincing and interesting piece for the community working on intracellular pathogens. The authors should also carefully re-edit their manuscript to avoid overselling their data (see below for issues I see there). I would consider taking out the first figure and starting with Figure 3 (Figure 2 could be re-organized in the later parts)- that could help to make the flow of the manuscript better.Strengths:The high-content analysis including the innovative analytical workflows are very promising and could be used by a large number of scientists working on intracellular bacteria. The finding that Septins (through SEPT7) are differentially regulated through actively secreting bacteria is very exciting and can steer novel research directions.

We thank the Reviewer for their constructive feedback and excitement for our results, including our findings on T3SS activity and *Shigella*-septin interactions. In accordance with the Reviewer’s comments, we avoid overselling our data in the revised version of the manuscript.

Weaknesses:The manuscript makes a connection between two research lines (1: Shigella infection and DNA/protein synthesis, 2: regulation of septins around invading Shigella) that are not fully developed - this makes it sometimes difficult to understand the take-home messages of the authors.

We agree that the manuscript is mostly technical and therefore some of our experimental observations would benefit from follow up mechanistic studies in the future. We highlight our vision for broader applicability in response to weaknesses raised by Reviewer 2.

It is not clear whether the analysis that was done on projected images actually reflects the phenotypes of the original 3D data. This issue needs to be carefully addressed.

We agree with the Reviewer that characterizing 3D data using 2D projected images has limitations.

We observe an increase in cell and nuclear surface that does not strictly imply a change in volume. This is why we measure Hoechst intensity in the nucleus using SUM-projection (as it can be used as a proxy of DNA content of the cell). However, we agree that future use of other markers (such as fluorescently labelled histones) would make our conclusions more robust.

Regarding the different orientation of intracellular bacteria, we agree that investigation of septin recruitment is more challenging when bacteria are placed perpendicular to the acquisition plane. In a first step, we trained a Convolutional Neural Network (CNN) using 2D data, as it is easier/faster to train and requires fewer annotated images. In doing so, we already managed to correctly identify 80% of *Shigella* interacting with septins, which enabled us to observe higher T3SS activity in this population. In future studies, we will maximize the 3D potential of our data and retrain a CNN that will allow more precise identification of Shigella-septin interactions and in depth characterization of volumetric parameters.

**Recommendations for the authors:**

**Reviewer #1 (Recommendations For The Authors):**
(1) To conclude that cell volume is indeed increased, the investigators should consider staining the cells with markers that demarcate cell boundaries and/or are confined to the cytosol, i.e., a cell tracker dye.

Staining using our SEPT7 antibody enables us to define cell boundaries for cellular area measurements (Novel Figure 1 - figure supplement 1A). However, we agree with the Reviewer that staining cells with additional markers (such as a cell tracker dye) would be required to conclude that cell volume is increased. We therefore adjust our claims in the main text (lines 107-115 and 235-246).

(2) Line 27: I understand what is meant by "recruited to actively pathogenic bacteria with increased T3SS activation." However, one could argue that there are many different roles of the intracytosolic bacteria in pathogenesis in terms of pathogenesis, not just actively secreting effectors.

T3SS secretion by cytosolic bacteria is tightly regulated and both T3SS states (active, inactive) likely contribute to the pathogenic lifestyle of *S. flexneri*. In agreement with this, we removed this statement from the manuscript (lines 27, 225 and 274).

(3) Line 88: Please clarify in the text that HeLa cells are being studied.

We explicitly mention that the epithelial cell line we study is HeLa in the main text (line 93), in addition to the Materials and methods (line 328).

(4) Line 97: is it possible to quantify the average distance of the nuclei from the cell perimeter? This would help provide some context as to what it means to be a certain distance from the nucleus, i.e., is there another way to point out that distance from nuclei correlates with movement inward post-invasion at the periphery?

To provide more context to the inward movement of bacteria to the cell centre, we provide calculations based on measurements in Figure 1G, I. If we approximate geometric shape of both cells and nucleus to a circle, the median radius of a HeLa cell is 31.1 µm^2^ (uninfected cell) and 36.3 µm^2^ (infected cell). Similarly, the median radius of the nucleus is 22.2 µm^2^ (uninfected cell) and 24.57 µm^2^ (infected cell).

However, we note that Figure 1F shows distance of bacteria to the centroid of the cell, which is the geometric centre of the cell, and which does not necessarily coincide with the geometric centre of the nucleus. We also note that nuclear area increases with infection (in a bacterial dose dependent manner). Finally, we note that these measurements are performed on max projections of 3D Z-stacks. In this case we cannot fully appreciate distance to the nucleus for bacteria located above it.

(5) Lines 212-213 - there is no Figure 9A, B - I think this should be Figure 7A, B.

Text has been updated (lines 216-217).

**Reviewer #2 (Recommendations For The Authors):**
Testing the analysis pipeline as a proof-of-concept question such as the comparison of caging around the laboratory strain as compared to one or a few clinical isolates or mutants of interest would help stress the relevance of this new, remarkable tool.

We thank the Reviewer for their enthusiasm.

Future research in the Mostowy lab will capitalise on the high-content tools generated here to explore the frequency and heterogeneity of septin cage entrapment for a wide variety of *S. flexneri* mutants and *Shigella* clinical isolates.

The sentence in line 215 ends with "in agreement with" followed by a reference.

Text has been updated (line 219).

The sentence in line 217 on the correlation between caging and T3SS is not very clear.

Text has been clarified (lines 221-223).

There is a typo in line 219 : "protrusSions"

Text has been updated (line 223).

**Reviewer #3 (Recommendations For The Authors):**
Major pointsThe quantitative analysis approach in Figure 1 has multiple issues. Some examples:(1) How was the cell area estimated? Normally, a marker for the whole cell (CellMask or similar) or cells expressing GFP would be good indicators. Here it is not clear to me what was done.

The cell area was estimated using SEPT7 antibody staining which is enriched under the cell cortex. CellProfiler was used to segment cells based on SEPT7 staining, using a propagation method from the identified nucleus based on Otsu thresholding. To provide more clarity on how this was performed, we now include a new figure (Figure 1- figure supplement 1A) showing a representative image of HeLa cells stained with SEPT7 and the corresponding cell segmentation performed with CellProfiler software, together with an updated figure legend explaining the procedure (lines 784–787).

(2) The authors use Hoechst and integrated z-projections (Figure 1 S1) as a proxy to estimate nuclear volume. Hoechst staining depends on the organization of the DNA within the nucleus and I find that the authors need to do better controls to estimate nuclear size - this would be possible with cells expressing fluorescently labeled histones, or even better with a fluorescently tagged nuclear pore/envelope marker. The current quantification approach is misleading.

We understand Reviewer #3’s concerns about using Hoechst staining as a proxy of nuclear volume, due to potential differences in DNA organisation within the nucleus.

Following the recommendation of Reviewer #3 in the following point 3, text has been updated (lines 107–115 and 235-246).

(3) Was cell density assessed for the measurements? If cells are confluent, bacteria could spread between cells within 3 hrs, if cells are less dense, this does not occur. When epithelial cells are infected for some hours, they have the tendency to round up a bit (and to appear thicker in z), but a bit smaller in xy. My suggestion to the authors (as they use these findings to follow up with experiments on the underlying processes) would be to tone down their statements - eg, Hoechst staining could be simply indicated as altered, but not put in a context of size (this would require substantial control experiments).

Local cell density was not directly measured, but the experiment was set up to infect at roughly 80% confluency (cells were seeded at 10^4^ cells/well 2 days prior to infection in a 96-well microplate, as described in the Materials and methods section) and to ensure bacterial spread between cells.

In agreement with Reviewer #3 we tone down statements in the main text (see response to point 2 above).

In addition, I found Figure 1 (and parts of Figure 2) disconnected from the rest of the manuscript, and it may even be an idea to take it out of the manuscript (that could also help to deal with my feedback relating to Figure 1). I would suggest starting the manuscript with the current Figure 3 and building the biological story with a stronger focus on SEPT7 (and its links with T3 secretion and actively pathogenic bacteria) from there on. As it stands, the two parts of the manuscript are not well connected.

We carefully considered this comment but following revisions we have not reorganised the manuscript. We believe that high-content characterisation of *S. flexneri* infection in Figure 1 and 2 provides insightful information about changes in host cells in response to infection. Following this, we move onto characterising intracellular bacteria (and in particular those entrapped in septin cages) in the second part of the manuscript (Figure 3-7). Similar methods were used to analyse both host and bacterial cells and results obtained offer complementary views on host-pathogen interactions.

My major reservation with the experimental work of the current version of the manuscript relates to Figure 5: The analysis of the septin phenotypes in Figure 5 seems to be problematic - to me, it appears that analysis and training were done on projected image stacks. As bacteria are rod-shaped their orientation in space has an enormous impact on how the septin signal appears in a projection - this can lead to wrong interpretation of the phenotypes. The authors need to do some quantitative controls analyzing their data in 3D. To be more clear: the example "tight" (second row) shows a bacterium that appears short. It may be that it's actually longer if one looks in 3D, and the septin signal could possibly fall in the category "rings" or even "two poles".

The deep learning training and subsequent analysis of septin-cage entrapment is done on projected Z-stacks, which presents limitations. Future work in the Mostowy lab will exploit this first study and dive deeper into 3D aspects of the data.

To address Reviewer #3’s concern, we include a sentence explaining that this analysis was performed using 2D max projections (lines 708 and 724), as well as acknowledging its limitations in the main text (lines 259-262).

Minor pointsThe scale bar in Fig 1 is very thin.

We corrected the scale bar in Fig. 1 to make it more visible.

Could it be that Figure 1F is swapped with Figure1E in the description?

Descriptions for Figure 1E and F are correct.

Line 27: what does "actively pathogenic bacteria" mean? I propose to change the term.

We agree with Reviewer #3 that “actively pathogenic bacteria” should be removed from the text. This update is also in agreement with Reviewer #1 (see Reviewer #1 point 2).

Line 28: "dynamics" can be confusing as it relates to dynamic events imaged by time-lapse.

Although we are making a snapshot of the infection process at 3 hpi, we capture asynchronous processes in both host and bacterial cells (eg. host cells infected with different bacterial loads, bacterial cells undergoing actin polymerisation or septin cage entrapment). We agree that we are not following dynamics of full events over time. However, our high content approach enables us to capture different stages of dynamic processes. To avoid confusion, we replace “dynamics” by “diverse interactions” (line 28), and we discuss the importance of follow-up studies studying microscopy timelapses (line 274).

Paragraph 59 following: the concept of heterogeneity was investigated in some detail for viral infection by the Pelkmans group (PMID: 19710653) using advanced image analysis tools. Advanced machine-learning-based analysis was then performed on Salmonella invasion by Voznica and colleagues (PMID: 29084895). It would be great to include these somewhat "old" works here as they really paved the way for high-content imaging, and the way analyses were performed then should be also discussed in light of how analyses can be performed now with the approaches developed by the authors.

We agree. These landmark studies have now been included in the main text (lines 71-74).

Line 181: I do not know what "morphological conformations" means, perhaps the authors can change the wording or clarify.

We substituted the phrase “morphological conformations” by “morphological patterns” to improve clarity in the main text (lines 185).

The authors claim (eg in the abstract) that they are measuring the dynamic infection process. To me, it appears that they look at one time-point, so no dynamic information can be extracted. I suggest that the authors tone down their claims.

Please note our response above (Minor points, Line 28) which also refers to this question.